# Genomics-Based Reconstruction and Predictive Profiling of Amino Acid Biosynthesis in the Human Gut Microbiome

**DOI:** 10.3390/microorganisms10040740

**Published:** 2022-03-30

**Authors:** German A. Ashniev, Sergey N. Petrov, Stanislav N. Iablokov, Dmitry A. Rodionov

**Affiliations:** 1A.A. Kharkevich Institute for Information Transmission Problems, Russian Academy of Sciences, 127994 Moscow, Russia; escobar.morente@gmail.com (G.A.A.); biology@iablokov.ru (S.N.I.); 2Vavilov Institute of General Genetics, Russian Academy of Sciences, 119991 Moscow, Russia; petrovsnwm@gmail.com; 3Sanford Burnham Prebys Medical Discovery Institute, La Jolla, CA 92037, USA

**Keywords:** amino acid metabolism, metabolic reconstruction, non-orthologous displacements, phenotype, metagenomics, human gut microbiome

## Abstract

The human gut microbiota (HGM) have an impact on host health and disease. Amino acids are building blocks of proteins and peptides, also serving as precursors of many essential metabolites including nucleotides, cofactors, etc. Many HGM community members are unable to synthesize some amino acids (auxotrophs), while other members possess complete biosynthetic pathways for these nutrients (prototrophs). Metabolite exchange between auxotrophs and prototrophs affects microbial community structure. Previous studies of amino acid biosynthetic phenotypes were limited to model species or narrow taxonomic groups of bacteria. We analyzed over 2800 genomes representing 823 cultured HGM species with the aim to reconstruct biosynthetic pathways for proteinogenic amino acids. The genome context analysis of incomplete pathway variants allowed us to identify new potential enzyme variants in amino acid biosynthetic pathways. We further classified the studied organisms with respect to their pathway variants and inferred their prototrophic vs. auxotrophic phenotypes. A cross-species comparison was applied to assess the extent of conservation of the assigned phenotypes at distinct taxonomic levels. The obtained reference collection of binary metabolic phenotypes was used for predictive metabolic profiling of HGM samples from several large metagenomic datasets. The established approach for metabolic phenotype profiling will be useful for prediction of overall metabolic properties, interactions, and responses of HGM microbiomes as a function of dietary variations, dysbiosis and other perturbations.

## 1. Introduction

The human gut microbiota (HGM) is a complex community of microorganisms, largely composed of anaerobic and facultative anaerobic bacteria that have a huge impact on human health and wellbeing [1,2,3]. The highest number of commensal bacteria is found in the large intestine (colon), followed by the small intestine. The intestinal microbiota participates in fermentation of dietary polysaccharides and proteins, synthesis of vitamins and other essential metabolites such as short-chain fatty acids (SCFAs), and the enzymatic transformation of endogenous metabolites such as bile acids [4]. Among other important functions, HGM produce microbial metabolites that mediate crosstalk between HGM and the immune system of the host, as well as playing an important role in the gut-brain axis [5,6,7]. 

Amino acids are major macronutrients in diets, serving as building blocks of proteins and peptides, as well as precursors of many essential metabolites including nucleotides and cofactors, both in the host and HGM metabolism [8]. Many HGM species have both amino acid biosynthetic and biodegradative capabilities. The human colon is characterized by a high abundance and diversity of protein degrading and amino acid utilizing microorganisms [4]. Diet-derived amino acids can be also directly incorporated into bacterial cells as building blocks for their synthesis of proteins or transformed to other essential metabolites. The unified catalog of HGM genes from over 200,000 metagenome-assembled genomes (MAGs) is enriched with functional categories related to amino acid transport and metabolism [9]. Many experimentally characterized HGM species possess complete biosynthetic pathways for all proteinogenic amino acids (prototrophs). However, other HGM species and strains lose the capability to synthesize de novo one or multiple essential amino acids, and thus require these nutrients for growth (auxotrophs). Previous studies have indicated that amino acid exchange between auxotrophs and prototrophs is an important factor shaping microbial community structure [10,11]. 

Previous in silico and in vitro studies of amino acid auxotrophies were limited to a handful of model microbial species or were conducted within a narrow taxonomic group such as enterobacteria [12]. The number of isolated and sequenced HGM microorganisms is rapidly growing [13,14,15,16]. Mapping of amino acid biosynthetic pathways and assignment of prototrophic and auxotrophic phenotypes for HGM genomes are prerequisite steps for predictive functional profiling of HGM communities. A subsystem-based comparative genomics approach implemented in the SEED genomic database and analysis platform [17] is widely used to capture, analyze, and extend biochemical pathways in diverse microorganisms with sequenced genomes. This platform is a public repository of thousands of microbial genomes automatically annotated using Rapid Annotation Subsystem Technology (RAST) [18], frequently expanded by newly sequenced genomes and providing a suite of expert tools for manual curation and metabolic reconstruction. This analysis includes the use of genomic context for the identification of candidate “missing genes” to fill-in gaps in known pathways [19] and the prediction of novel pathway variants and corresponding phenotypes [20]. SEED subsystems allowed us and others to capture many aspects of microbial metabolism including carbohydrate utilization [21,22], and the biosynthesis of amino acids [23,24], vitamins and cofactors [20]. 

Recently, we applied the subsystem-based metabolic reconstruction to infer biosynthetic and salvage pathways for eight B-vitamins and queuosine in a reference set of 2228 HGM genomes [25]. Based on these in silico reconstructions, we developed *phenotype rules* and assigned simplified binary (“1” or “0”) phenotypes describing corresponding vitamin prototrophy and auxotrophy in these reference genomes. The predicted vitamin binary phenotypes combined into a Binary Phenotype Matrix (BPM) were further used for predictive phenotype profiling from 16S rRNA gene-based phylogenetic profiles representing a large HGM dataset from the American Gut Project (AGP). As a result, for each metagenomic sample and each phenotype, we computed a Community Phenotype Index (CPI) representing the expected fraction of cells possessing a particular phenotype. This in silico analyses revealed common trends for certain vitamins, such as the highest mean CPI values for B2, B3, B6 and B9 and lowest CPIs for B7 [25]. In a follow-up study, it was applied in conjunction with in vitro co-culturing and experiments in HGM-colonized gnotobiotic mice to assess the impact of dietary B-vitamins [26]. Predictive functional profiling combined with dietary variations and multi-omic measurements in colonized gnotobiotic mice was effectively applied to the analyses of defined microbial consortia of 10–30 HGM strains representing different stages of microbiota succession in infants displaying normal and pathological (resulting from acute malnutrition) growth and development [27,28,29,30]; this was also shown to be useful for microbiota-based diagnostics of medical conditions such as IBD [31].

Here, we applied the subsystem approach and predictive phenotype profiling to analyze the distribution of amino acid metabolism genes in an expanded collection of 2856 bacterial genomes representing over 800 distinct HGM species. Using genomic-based metabolic reconstruction, we systematically mapped 163 enzymatic and 47 transporter components involved in the metabolism of 19 amino acids in this genomic collection. We report non-orthologous replacements of genes encoding core enzymes in serine, arginine, lysine, and threonine biosynthesis. The reconstructed pathways allowed us to classify the studied organisms with respect to their biosynthetic and transport capabilities and determine variability of the phenotypic profiles at different taxonomy levels. The obtained amino acid auxotrophy phenotypes were compared with published experimental data on nutritional requirements of HGM bacteria. Finally, we investigated the cumulative phenotypic properties of human stool samples using 16S rRNA gene sequencing and shotgun metagenomic data from several major HGM studies.

## 2. Materials and Methods

### 2.1. HGM Genomic Collection

A set of genomes representing HGM bacteria was selected using the approach previously described in [25]. Mapping of the list of 823 selected HGM species to the PATRIC genomic database [32] was conducted in April 2020. The obtained list of 2856 bacterial strains was represented by either complete or high-quality draft genomes imported to the SEED database for further annotation and metabolic reconstruction (Appendix A). Phylogeny analysis was held by generating a maximum likelihood phylogenetic tree based on concatenated multiple alignments of eleven ribosomal proteins extracted from the analyzed set of genomes in the SEED database [18]. Multiple alignments of ribosomal proteins were obtained using MUSCLE version 3.8.31 [33]. The phylogenetic tree was constructed by RAxML version 8 [34] and visualized using the iTOL web server [35] with taxonomic assignments obtained from the NCBI Taxonomy Database (Appendix A). The selected reference set of genomes included 2578 genomes with taxonomic assignments to the species level, while 241 and 28 genomes have taxonomy assigned up to the genus and family levels, respectively.

### 2.2. Metabolic Reconstruction

For genomic reconstruction of amino acid biosynthetic pathways and metabolic phenotype prediction across the list of specified 2856 genomes, we used a subsystem-based approach implemented in the SEED genomic platform [17,18]. This approach is based on functional gene annotation and prediction using two comparative genomics techniques: (i) homology-based methods and (ii) genome context analysis. Analyses of gene co-occurrence, co-regulation and clustering on the bacterial chromosome were three main genome context analysis techniques used for functional gene annotation and metabolic reconstruction [36]. The SEED subsystems are sets of functional roles that capture current pathway knowledge across the analyzed genomes [37]. Each functional role corresponds to a set of homologous genes that implement this role in a specific subset of organisms (Appendix A). First, we scanned the analyzed proteomes against the KEGG Orthology [38], TCDB [39], UniProtKB [40] and SEED [17] databases to reveal proteins potentially involved in the amino acid metabolism and transport, with additional functional annotations obtained by literature searches using the PaperBLAST tool [41]. Then, we analyzed the genomic and functional contexts of gene loci encoding the obtained proteins and reconstructed the respective metabolic pathways and transcriptional regulons. For the comparative genomics-enabled pathway and regulon inference, we used additional closely related bacterial genomes available in the SEED database. Transcriptional regulons (sets of genes co-regulated via a shared transcription factor or shared RNA elements, riboswitches, responding to respective pathway metabolites) were predicted and reconstructed using the comparative genomics approach [42] implemented in the RegPrecise database [43]. This integrative subsystems-based approach was previously used for the reconstruction of pathways and regulons involved in carbohydrate utilization, amino acid biosynthesis and vitamin metabolism in the *Shewanella*, *Bacteroides*, and other microbial lineages [21,22,23,25,44,45,46].

The final reconstructed metabolic pathways for all proteinogenic amino acids, except alanine, contained 283 functional roles including 163 enzymes, 51 transporter components and 11 transcriptional regulators (Appendix A). These include alternative biochemical pathways involved in the synthesis of chorismate (a common metabolic precursor of aromatic amino acids), glutamate, glutamine, asparagine, lysine, methionine, and glycine (Figure 1). Moreover, 32 biochemical reactions (corresponding to 103 functional roles) were represented by two or more alternative isozymes encoded by non-orthologous genes. These non-orthologous gene displacements were predicted based on biochemical and functional identity of gene candidates, their genomic co-localization, co-occurrence, and co-regulation along with the reconstructed pathway context.

### 2.3. Functional Profiling of 16S Metagenomics Datasets

For each analyzed metabolic pathway, we further classified genomes by patterns of occurrence of signature pathway genes and assigned them pathway variants and growth requirements (Table 1). Using phenotype rules, these pathway variants were translated to either prototrophic or auxotrophic phenotypes, expressed as binary phenotypes, “1” and “0”, respectively. The obtained phenotypes in a binary form comprise a binary phenotype matrix (BPM), which was further used for the quantitative analysis and predictive profiling of amino acid biosynthetic capabilities in HGM metagenomic samples as previously described [25,47]. Briefly, raw 16S rRNA gene sequencing data from two large metagenomic studies of urban HGM of western cohorts, the AGP (including 2868 samples) [48] and the UK twins (UKT, 3288 samples) [49], were quality-filtered and dereplicated into amplicon sequence variants (ASVs) using the QIIME2 pipeline [50]. For the taxonomic classification of ASVs we used the Multi-Taxonomic Assignment (MTA) approach [47] and the union of the NCBI 16S and RDP databases. We used the same approach to analyze 16S samples from the Hadza dataset representing 333 HGM samples collected from a rural community of the Hadza hunter-gatherer people from northern Tanzania [51]. 

To assess the amino acid biosynthesis potential of the selected HGM metagenomic samples, we used a development version of the Phenobiome Profiler tool provided by PhenoBiome Inc. (Walnut Creek, CA, USA). The pipeline consists of several steps, first of which establishes a map between the analyzed ASVs and the reference organisms in the BPM based on the 16S rRNA nucleotide identity (for details see [47]). ASVs with high nucleotide identity (greater than 0.9) were considered “mapped”, while the other ASVs were considered as “non-mapped” and discarded. Samples with less than 75% abundance coverage (i.e., the abundance of “mapped” ASVs) were discarded, resulting in 2130 (AGP), 2679 (UKT), and 145 (Hadza) samples retained for further analysis. Next, this map was used to assign Phenotype Indices (PI), i.e., probabilistic estimates (on the scale from 0 to 1) for a given ASV to be a particular phenotype carrier (e.g., an amino acid prototroph). Finally, a cumulative characteristic CPI was calculated as an abundance-weighted average PI for each amino acid. Additionally, we used the Faith alpha-diversity metric to evaluate Phenotype Alpha Diversity (PAD) for each phenotype [47]. Thus, PAD_1 and PAD_0 correspond to the alpha diversities for the sub-communities of phenotype carriers and non-carriers, respectively.

### 2.4. Comparison of Predicted Functional Profiles with the State-of-the-Art PICRUSt2 Approach

To compare the CPI-based phenotype profiling approach with a state of the art predictive metabolic pathway abundance approach [52], we used ASV sequences and abundance tables obtained for the AGP dataset and run PICRUSt2 with default parameters [53]. The default use case for PICRUSt2 allows one to predict: (i) the abundance of KEGG ortholog (KO) families, and (ii) the abundance of known metabolic pathways from the MetaCyc database [54] using KO functional annotations and the Minimal Set of Pathways (MinPath) algorithm [55]. The predicted abundances of selected amino acid synthesis pathways in MetaCyc were normalized by read numbers in each sample. We further used the PICRUSt2 algorithm to predict the abundances of binary metabolic phenotypes in the AGP datasets using the obtained BPM for amino acid production in 2856 reference genomes. First, we mapped genomes from the PICRUSt2 reference tree to the BPM genomes using their NCBI TaxIDs and, thus, prepared a custom traits table for 2607 leaves. Then, we used this BPM-based trait table with the PICRUSt2 pipeline to calculate cumulative phenotype abundances in 16S samples and, finally, normalize them by read number to get the relative phenotype abundance (RPA) values.

### 2.5. Functional Profiling of Shotgun Metagenomes

We analyzed whole genome shotgun sequencing (WGS) metagenomic samples from the Integrative Human Microbiome Project (iHMP) on functional dysbiosis in the gut microbiome during Inflammatory Bowel Disease (IBD) activity [56]. The analyzed 384 WGS fastq files were filtered to remove host-specific reads using Bowtie2 [57], and the hg38 human genome assembly. We further performed quality filtering of WGS reads using the KneadData package (KneadData—The Huttenhower Lab) to enable adapter removal, trimming and filtering by quality. To obtain taxonomic profiles of the WGS samples we used Kraken 2 and Bracken [58] and our custom genomic database containing 2856 reference HGM bacteria with taxonomic assignments according to the NCBI Taxonomy Database. To assess the metabolic potential for amino acid production in the WGS samples, we used the Phenotype Profiler tool with the relative taxonomy abundance profiles provided as an input and a taxonomy-based approach to map the respective taxonomic assignments to the reference BPM organisms (for details, see [47]). The WGS entries with taxonomic similarity to the BPM worse than on a family level were marked as “non-mapped” and discarded, with the respective relative abundances of “mapped” entries renormalized to sum to 1. As a result, the predicted metabolic phenotype profiles included CPI values calculated for each WGS sample and each analyzed amino acid phenotype. 

For the gene-based functional profiling of trimmed and filtered WGS data files we implemented a pipeline including the following public domain tools: a metagenome assembly with MEGAHIT [59]; gene prediction with Prokka (v1.14, metagenomic mode) [60]; functional annotation by a protein similarity search with DIAMOND [61]; and mapping of WGS reads to the functionally annotated genes using Bowtie2 [57]. For functional annotation we used complete proteomes of 2856 reference HGM genomes that include both functionally annotated proteins from the reconstructed metabolic pathways and representative sequences of all other proteins from these genomes. Finally, we sum up the number of mapped reads for genes with the same functional role from the reconstructed metabolic pathways using bedtools [62]. At the final step, we performed gene count normalization using the Trimmed Mean of M-values (TMM) approach [63] implemented in the edgeR package [64]. For TMM-normalization, we used a core gene set that is a set of universal single-copy genes that are present in all genomes in our reference database. The gene count matrix included only genes that either belong to a set of the functionally annotated genes from the studied amino acid biosynthetic pathways or genes from the core gene set. As a result, the predicted functional gene profiles included the TMM-normalized total abundances of genes encoding pathway-specific reactions in each amino acid production pathway.

## 3. Results

### 3.1. Genomic Reconstruction of Amino Acid Biosynthetic Pathways

In order to reconstruct amino acid metabolism in HGM communities, we first compiled the list of 2856 representative bacterial genomes using the MetaHIT consortium [65], the Human Microbiome Project (HMP) [66], and literature [13], and further mapped the obtained HGM species onto the PATRIC genomic database [32]. The obtained HGM genomic collection was represented by 823 taxonomic species, 296 genera, 104 families, 42 orders, 23 classes and 11 phyla (Appendix A). The majority of the analyzed genomes belong to the Firmicutes (47%), Proteobacteria (22%), Actinobacteria (18%) and Bacteroides (11%) phyla. The *Bifidobacterium* and *Streptococcus* genera included over 200 genomes, while each of the *Bacillus*, *Bacteroides*, *Enterococcus*, *Escherichia*, *Lactobacillus*, *Prevotella*, and *Staphylococcus* genera were represented by 100–200 genomes.

For reconstruction of the metabolic pathways of amino acid biosynthesis and the identification of uptake transporters we applied the metabolic subsystem approach implemented in the SEED genomic platform and database [18] using the comparative genomics workflow described in our previous analysis of vitamin metabolic pathways [25]. For every proteinogenic amino acid, we created a metabolic subsystem table populated by the occurrence of specific functional roles in the analyzed set of bacterial genomes (Appendix A). Alanine was excluded from this analysis as it is ubiquitously produced by bacteria in a single biochemical step of pyruvate transamination, and therefore does not imply a biochemical pathway. Moreover, we reconstructed a biosynthetic pathway for chorismate, which is a common precursor of three aromatic amino acids (Trp, Tyr, Phe). Three branched chain amino acids (Ile, Leu, Val), two aromatic amino acids (Phe, Tyr), as well as the two pairs of related amino acids, Asn and Asp, and Gln and Glu, were combined together into the respective subsystems (Table 1 and Figure 1). The developed subsystems included the functional roles of previously known enzymes and transporters from literature, reference metabolic pathway databases including KEGG [38] and MetaCyc [54], and also 7 novel enzymes predicted to be involved in amino acid metabolism as non-orthologous gene displacements (see below).

As a result, 15 metabolic subsystems were populated by 213 functional roles including 155 enzymes catalysing 99 distinct biochemical reactions (each corresponds to a unque Enzyme Commission (EC) number), 47 components of 23 amino acid uptake transporters and 3 transcriptional regulators for amino acid metabolism (Appendix A). The reconstructed amino acids biosynthetic pathways included several alternative pathways variants (Table 1 and Figure 1) including: (i) two variants for branched chain amino acids that have the same core pathway and differ in the upstream source of 2-oxobutanoate; (ii) four variants of the lysine biosynthesis pathway that differ by the intermediate route of conversion of tetrahydrodipicolinate to meso-2,6-diaminopimelate; (iii) two methionine synthesis pathway variants that differ by the source of sulfur (cysteine or hydrogen sulfide); (iv) two routes for glycine biosynthesis (staring from threonine or serine); (v) two alternative upstream pathways in the chorismate biosynthesis; (vii) two alterantive enzymes for synthesis of glutamate from oxoglutarate (glutamate synthase GltBD and glutamate dehydrogenase Gdh), and asparagine from aspartate (aspartate–ammonia ligase AsnA and asparagine synthetase AsnB). Furthermore, in the glutamine metabolic subsystem, we included an alternative route represented by glutamyl-tRNA amidotransferase (GatABC), which provides a means of producing correctly charged Gln-tRNA(Gln) through the transamidation of mis-acylated Glu-tRNA(Gln) in organisms which lack glutaminyl-tRNA synthetase [67]. The same GatABC amidotransferase was also added to the asparagine subsystem, since it was shown that this enzyme is equally efficient in the transamidation of Asp-tRNA(Asn) and Glu-tRNA(Gln), thus it can compensate for the absence of AsnA/AsnB in *Helicobacter pylori*, *Acinetobacter baumannii*, *Bifidobacterium longum*, *Staphylococcus aureus* and many other microorganisms [68] (Table 1).

### 3.2. Non-Orthologous Gene Displacements in Biosynthetic Pathways

The reconstructed amino acid biosynthetic pathways include 103 functional roles for enzymes represented by non-orthologous gene displacements (NODs) that catalyze 33 biochemical reactions in total (Appendix A). These include seven new NOD enzymes from the arginine, serine, lysine, and threonine biosynthetic pathways (Table 2). Functional and comparative genomic analysis of these novel NODs in the amino acid biosynthesis of HGM species is described below.

In the arginine biosynthetic pathway, we identified two novel NODs for N-acetylglutamate synthase ArgA, which catalyzes the first step in the pathway, the conversion of L-glutamate to N-acetyl-L-glutamate (Figure 1). ArgA is missing in 228 genomes of HGM bacteria possessing an incomplete pathway variant. The majority of these species belong to the Bacteroidetes phylum (194 genomes). In all these genomes, we identified a novel candidate enzyme (named ArgA2) previously annotated as a putative GNAT-family N-acetyltransferase in bacteria from the *Bacteroides*, *Prevotella*, *Alistipes*, and other genera, which is encoded by a gene located within their arginine operons (Figure 2). An alternative arginine biosynthetic pathway involving succinyl derivatives was previously characterized in Bacteroides fragilis [69], where a novel N-succinyl-L-ornithine carbamoyltransferase (ArgF3) replaces the classic ArgF enzyme (e.g., from *E. coli*), which is specific for N-acetyl-L-ornithine [70]. It was also shown that the second step of arginine biosynthesis in *B. fragilis* is catalyzed by N-succinyl-L-glutamate kinase [69], which is an ortholog of ArgB, thus suggesting the use of succinylated intermediates for arginine synthesis. Interestingly, the ArgF3 proteins from *Bacteroides* spp. showed low sequence similarity to the *E. coli* ArgF protein (~26% identity) and revealed a characteristic for ArgF3 proline-rich loop, which is absent from ArgF proteins. We revealed a co-occurrence of ArgF3 and ArgA2 in 190 *Bacteroidetes* genomes, moreover, the *argA2* genes are colocalized with *argF3* in 23 of these genomes. Thus, we proposed N-succinyl-L-glutamate synthase function to ArgA2 in the Bacteroidetes phylum. This hypothesis on involvement of ArgA2 in arginine biosynthesis was recently confirmed by large-scale genetic data in *B. thetaiotaomicron*, which shows that the ArgA2-encoding gene BT3761 is critical for growth on minimal media without amino acids, and its fitness defect was rescued through the addition of arginine [71]. Finally, we detected another NOD for a missing ArgA enzyme in 12 *Vellionela* and *Dialister* genomes, named ArgA3, which is homologous to N-acetylcysteine deacetylase from *Bacillus subtilis* (Uniprot ID P54955, 39% identity). The genomic context of *argA3*, namely its co-occurrence and colocalization with other *arg* genes, confirms its functional involvement in arginine biosynthesis (Figure 2).

The serine biosynthetic pathway contains three NODs for two major enzymes, SerC and SerB, that catalyze the transamination of phosphohydroxypyruvate to phosphoserine and its subsequent dephosphorylation to serine, respectively (Figure 1). The previously known type of phosphoserine aminotransferase (SerC1) was absent in 217 HGM genomes possessing two other serine biosynthetic enzymes, SerB and SerA. Among these genomes, we identified a novel NOD of aminotransferase, named SerC2, which form the same operon with SerA and SerB in 113 genomes from the *Staphylococcus* genus. Another potential phosphoserine aminotransferase isozyme, named SerC3, was identified in 173 genomes including 120 genomes that lack both *s**erC1* and *serC2* genes. The genome context analysis suggests that *serC3* is located within the same putative operon with *serA* in 65 genomes including the *Clostridium*, *Akkermansia* spp., and *Brevibacillus* spp., while *Brachyspira pilosicoli* has all three serine biosynthesis genes organized into a single *serC3-serA-serB1* operon (Figure 2). Both SerC2 and SerC3 belong to the same family of aminotransferases (Pfam ID PF00266) as SerC1 and are similar to a recently characterized 3-phosphoserine transaminase from *Synechocystis* sp. (Uniprot ID P74281) [72], thus supporting the predicted biochemical function of these NODs.

The last step of serine biosynthesis is catalysed by phosphoserine phosphatase SerB, named SerB1 in our study, which was characterized in *E. coli* [73] and many other bacterial species [74]. SerB1 is absent in 660 HGM genomes encoding two other serine pathway enzymes, SerA and SerC. Alternative YseA enzyme with phosphoserine phosphatase activity (termed SerB3 here) was identified in *Bacillus subtilis* by screening against a large library of phosphorylated compounds [74]. Here, we tentatively identified two other phosphoserine phosphatase NODs: (i) SerB2 in 58 genomes; and (ii) SerB4 in 120 genomes (Figure 2). The SerB2 isozyme belongs to the histidine phosphatase superfamily (Pfam ID PF00300), while SerB4 is a putative phosphatase from the HAD superfamily, which has an additional N-terminal domain annotated as the haloacid dehalogenase-like hydrolase (Pfam ID PF00702). SerB2 shows 32% identity to recently characterized phosphoserine phosphatase from *Hydrogenobacter thermophilus* [75]. Genomic co-localizations of the *serB2* and *serA* genes in *Peptoniphilus* spp., as well as *serB4* with the *serC3* and *serA* genes in *Brevibacillus* spp., *Bacillus fordii*, and *Staphylococcus* spp. suggest an involvement of these novel phosphatases in the serine biosynthesis pathway.

The threonine biosynthetic pathway requires two specifc enzymes, homoserine kinase ThrB and threonine synthase ThrC, and also involves homoserine dehydrogenase Hom, which is shared with the methionine biosynthesis pathway (Figure 1). ThrB is absent in 328 out of 2533 predicted threonine prototrophs among the analyzed HGM genomes. A novel predicted functional analog of homoserine kinase, named ThrB2, was identified in 300 HGM genomes, two thirds of which are members of the Bacteroidetes phylum, while the remaining genomes belong to the Firmicutes phylum. These include 278 genomes with missing *thrB* gene, thus *thrB* and *thrB2* demonstrate complementary genomic occurrence profiles. In the majority of Bacteroidetes species, the *thrB2* gene is a part of a three-gene operon encoding ThrC and Asd-Hom, a bifunctional enzyme involved in homoserine synthesis (Figure 2). We also identified a few other Bacteroidetes species (e.g., *B. propionicifaciens* and *B. coprosuis*) that possess similar threonine biosynthesis operon, where *thrB2* is substituted with *thrB*, suggesting these two genes encode interchangeable enzymes. Among Firmicutes, *thrB2* is co-localized with the *asd* and *hom* genes in nearly 50 out of 100 HGM genomes, including *Blautia*, *Clostridium*, *Eubacterium*, and *Roseburia* spp. Many of the identified *thrB2* genes are currently annotated as a putative phosphoglycerate mutase because their protein products are ~35% identical to 2,3-bisphosphoglycerate-independent phosphoglycerate mutase (PGM) from Archaea (Uniprot ID O57742), which has a metalloenzyme domain fused to the PGM catalytic domain and is dependent on the presence of metal cations [76]. Interestingly, a large-scale chemical-genetics screening of genes recently identified the ThrB2-encoding BT2402 gene as a potential replacement of ThrB in *B. thetaiotaomicron* as it was required for growth in minimal medium, while the addition of threonine has rescued this growth deficiency [77]. In summary, we conclude that ThrB2 is a promising candidate for NOD of a traditional homoserine kinase in many HGM species.

Lysine biosynthesis is represented by four pathway variants, eleven biochemical reactions and sixteen dedicated enzymes including three alternative forms of L,L-diaminopimelate aminotransferase (DapL), two isozymes of N-succinyl-L,L-diaminopimelate aminotransferase (DapC), and two variants of diaminopimelate epimerase (DapF), which synthesizes meso-2,6-diaminopimelate from LL-2,6-diaminopimelate (Figure 1, Appendix A). A novel diaminopimelate epimerase isozyme, named DapF2, belongs to the family of pyridoxal phosphate dependent isomerases, specifically those racemases and epimerases acting on amino acids and derivatives. DapF2 is homologous to alanine racemase from *Corynebacterium glutamicum* (Uniprot ID Q8RSU9, 29% similarity) and lysine racemase from *Oenococcus oeni* (Uniprot ID Q04HB7 27% similarity). The *dapF2* gene was found in 125 HGM genomes possessing the incomplete lysine biosynthetic pathway with missing *dapF*. The majority of DapF2-encoding genomes are from the *Staphylococcus* genus, namely *S. aureus*, *S. epidermidis*, *S. hominis*, etc. (114 genomes), and also include a small group of *Gemella* sp. (6 genomes) and *Peptostreptococcus* sp. (5 genomes). Based on this gene occurrence pattern, genomic co-localization of *dapF2* with other lysine biosynthetic genes including *lysA*, *dapB*, *dapA*, and *dapH* (Figure 2) and their co-regulation by the lysisne riboswitch [24], we propose that DapF2 is a NOD for a traditional form of diaminopimelate epimerase DapF.

### 3.3. Incomplete Pathway Variants and Salvage of Amino Acid Precursors

Metabolic reconstruction of amino acid biosynthesis in 2856 microbial genomes revealed the presence of incomplete pathways (Table 1). First, along with the canonical biosynthetic pathway variants (denoted as P, P1, P2, etc.), we introduced P* variants with missing enzymes that are potentially substituted by yet unknown alternative enzymes. The threonine synthesis pathway is incomplete in 50 bacterial genomes (mostly Clostridia) that lack both homoserine kinase isozymes (ThrB and ThrB2). The serine pathway lacks all four alternative phosphoserine phosphatases in 450 genomes representing diverse Firmicutes species. The branched-chain amino acid (BCAA) biosynthetic pathway in 17 genomes (mostly Actinobacteria) lacks both alternative forms for the first enzymatic step, namely threonine dehydratase (IlvA) and citramalate synthase (CimA). In the lysine biosynthetic pathway, there are 38 genomes mostly from the *Alistipes* and *Capnocytophaga* genera with missing amination pathway enzymes converting tetrahydrodipicolinate to LL-2,6-diaminopimelate but, at the same time, that possess all other enzymes essential for lysine formation, namely DapA, DapB, DapF, and LysA (Figure 1). We hypothesized that all of the above-described P* variants correspond to respective amino acid prototrophic phenotypes and that future studies will help to identify missing biosynthetic enzymes or metabolic routes.

On the other hand, we also identified incomplete pathway variants with missing key biosynthetic enzymes, and the corresponding genomes were assigned amino acid auxotrophic phenotypes (denoted as A*, A1, A2, A3, etc.). For example, the arginine pathway in 251 genomes is represented by only 2 out of 8 enzymatic steps, namely argininosuccinate synthase (ArgG) and argininosuccinate lyase (ArgH), that are involved in the formation of arginine from citruline (variant A1). On the contrary, we also identified five arginine auxotrophs that are capable of synthesizing citrulline but lack ArgG and ArgH (variant A2), suggesting that citrulline could be involved in metabolic exchange in HGM communities. The BCAA biosynthetic pathways are incomplete in 20 HGM genomes (such as *Lactobacillus* and *Sutterella* spp.), possessing only the leucine biosynthetic genes with apparently missing upstream pathways for isoleucine and valine biosynthesis (variant A*). In the methionine pathway, the upstream part of the pathway leading to homocysteine from homoserine is absent in 83 HGM genomes (variant A1), while both methionine synthase isozymes (MetH and MetE) are missing in 25 genomes (variant A2). The lysine pathway in 14 genomes lacks all enzymes leading to meso-2,6-diaminopimelate formation (variant A1), while the presence of the last enzymatic step (LysA) in these HGM species suggests their potential capability to salvage the diaminopimelate precursor of lysine. Three incomplete pathway variants were found for tryptophan biosynthesis. The A1 variant assigned to 14 HGM strains includes a single enzyme, tryptophan synthase (TrpAB), that catalizes the final step of the pathway—condensation of indole and serine to tryptophan. The A2 variant identified in 82 *Propionibacterium acnes* strains, three *Clostridium* and two *Streptoccocus* spp. genomes contains one additional enzyme, TrpC, catalyzing an upstream reaction leading to the formation of 3-indoyl-glycerolphosphate. The remaining group of predicted tryptophan auxotrophs (A3 variant, 38 genomes) are not able to synthesize anthranilate due to the absence of TrpEG; however, they possess all of the downstream enzymes for tryptophan synthessis from anthranilate. Finally, we identified a group of eleven *Lactobacillus* and two *Anaerotruncus* strains that have an incomplete pathway of chorismate biosynthesis represented by AroK, AroA, and AroC enzymes (Figure 1), allowing these auxotrophic species to use shikimate for chorismate synthesis (variant As). In general, the conducted analysis of incomplete amino acid biosynthesis pathways identified potential metabolic crosstalks among HMG species.

### 3.4. Predicted Amino Acid Synthesis Phenotypes and Growth Requirements

Based on the obtained distribution of amino acid biosynthetic enzymes and pathways among 2856 HGM genomes, we assigned prototrophic and auxotrophic pathway variants and predicted their growth requirements (Table 1). Since biosynthetic pathways for several amino acids are mutually dependent on other amino acids that serve as essential metabolic precursors (Figure 1), we analyzed pathway dependencies characterized by a combination of auxotrophic and prototrophic phenotypes for metabolically interconnected amino acids in more detail. For cysteine and tryptophan, both amino acid biosynthetic pathways require serine as a precursor, we identified their dependencies on serine in 411 and 140 genomes, respectively. These include many *Streptococcus* and *Veillonella* strains that are predicted serine auxotrophs with complete pathways for cysteine and tryptophan. Glycine is synthesized from either serine or threonine by single-step pathways; we thus identified 395 genomes possessing the serine to glycine pathway (GlyA or SgaA) but lacking serine biosynthesis, suggesting their growth requirement for either glycine, or serine (P1 phenotype). In contrast, only 25 genomes with predicted threonine auxotrophy possess the threonine to glycine pathway (GlyB) as the only glycine biosynthetic pathway (P2 phenotype). Another 23 genomes possess both glycine biosynthetic pathways but lack both threonine and serine biosynthesis, suggesting their growth requirement for either threonine, serine, or glycine (P3 phenotype). Biosynthesis of all three aromatic amino acids (Phe, Trp, Tyr) requires a common metabolic precursor, chorismate, which is also used for synthesis of the folate precursor para-aminobenzoate. Among the analyzed HGM genomes, we identified 15 chorismate auxotrophic strains that possess at least one downstream pathway for aromatic amino acid synthesis, including five *Lactobacillus fermentum* and three *Dialister invisus* strains. The BCAA biosynthetic pathway that utilizes threonine as a precursor (signature enzyme IlvA) was identified in 20 threonine auxotrophic strains including all six *Akkermansia* genomes. Glutamate serves as a precursor for arginine, proline, and aspartate biosynthesis. In our genomic collection, we found 14, 36 and 113 genomes that lack glutamate synthesis but possess pathways for the biosynthesis of arginine, proline, and aspartate, respectively. Finally, we found 19 Asp-auxotrophic strains (e.g., *Helicobacter pylori*) that require Asp for their lysine biosynthesis. Overall, the predicted mutual dependencies of amino acid biosynthetic pathways suggest an expanded role of metabolite exchange between auxotrophic and prototrophic HGM strains.

To validate the predicted amino acid synthesis phenotypes, we searched published experimental data on amino acid auxotrophies for each of 800 analyzed HGM species. Analysis of amino acid growth requirements in 33 bacterial species revealed that 117 out of 135 experimentally determined amino acid requirement phenotypes (87%) agreed with the predicted amino acid auxotrophies (Appendix A). For instance, cysteine growth requirement was previously established for a large number of *Bifidobacterium* species and strains including 11 *Bifidobacterium* species analyzed in this work that all lack the cysteine biosynthetic pathway, with only the exception of *B. scardovii*, which was capable of growing slowly in the Cys-deficient medium [78]. A few cases of inconsistency for Cys and Met phenotypes in *Lactobacillus* and *Listeria* spp. could be explained by the absence of enzymes and/or transporters providing hydrogen sulfide, an essential precursor for both sulfur-containing amino acids. The remaining inconsistencies could possibly occur due to alternative biosynthetic enzymes (for Glu, Asp in *Lactobacillus* spp.), or explained by strain-specific phenotype variations, when the experimentally assessed strain may have phenotypes distinct from the reference strains analyzed in this work (such as the Pro and Ser auxotrophies in *Clostridium sporogenes*). We further analyzed the experimentally characterized amino acid production capabilities of HGM species and found a perfect correlation of these phenotypes with those predicted from current genomic analyses amino acid prototrophic phenotypes (Appendix A). In summary, the examined experimental data on amino acid requirements and production corresponds well to our in silico reconstruction and prediction of amino acid biosynthesis phenotypes.

### 3.5. Phylogenetic Variability of Binary Amino Acid Synthesis Phenotypes

The obtained pathway variants for amino acid prototrophs and auxotrophs were translated to binary (“1” and “0”) phenotypes (Table 1). Next, we calculated the averaged amino acid prototrophic phenotype values at the level of species, genus, family, order, class, and phylum (Appendix A). Phylogenetic distribution of the obtained averaged phenotype values demonstrates that the majority of HGM genera from Actinobacteria, Bacteroidetes, Firmicutes and Proteobacteria phyla are prototrophs for most amino acids (Figure 3). For instance, among Actinobacteria, all 242 analyzed genomes of *Bifidobacterium* spp. can synthesize all amino acids except cysteine, while 84 *Cutibacterium* spp. are auxotrophs for methionine, phenylalanine, tryptophan and BCAA. Most Fusobacteria are auxotrophs for all amino acids except asparagine, aspartate, glutamine, glutamate, while the Tenericutes phylum mostly includes auxotrophic strains for all amino acids. The Lentisphaerae and Planctomycetes phyla, each represented by a single genome, are prototrophic for all amino acids, except auxotrophy for serine and cysteine in *Victivallis vadensis*. Spirochaetes (2 genomes) are auxotrophic for arginine, histidine, and tryptophan, while 3 analyzed strains of Synergistetes show mixed patterns of prototrophy for 10 amino acids. Finally, the Verrucomicrobia strains are prototrophic for all amino acids except threonine (Figure 3B). 

To enable the accurate phylogeny-based projection of amino acid synthesis phenotypes from reference HGM genomes to phylotypes from metagenomic samples we assessed binary phenotype variations at various taxonomic levels using two metrics: (i) number of variable phenotypes (NVP), and (ii) overall phenotype variability score (OPVS), which was calculated as a sum of variances for each amino acid phenotype, as previously described [25] (Appendix A). For nineteen amino acid synthesis phenotypes, the cumulative OPVS metric ranges between 0 and 9.5. Among 830 analyzed HGM species, 330 species are represented by two or more strains, including 213 species containing three or more strains (Appendix A). Of the latter group, 55 species have at least one variable amino acid phenotype (NVP > 0). Five *Lactobacillus* spp., *Campylobacter jejuni*, *Clostridium botulinum*, *Fusobacterium nucletum*, *Kytococcus sedentarius*, and *Sutterella wadsworthensis* showed the highest NVP and OPVS values at the species level. *Lactobacillus brevis,* represented by three analyzed HGM genomes, shows the highest variability of the amino acid synthesis phenotypes (NVP = 15, OPVS = 5), suggesting that one of these strains could be incorrectly classified. Indeed, *L. brevis* subsp. *gravesensis* ATCC 27305, which is a predicted prototroph for all amino acids except cysteine, while two other *L. brevis* strains are auxotrophs for all but three amino acids, was recently re-classified as *Lactobacillus hilgardii* strain. 

At the genus level, 163 out of 296 analyzed genera are represented by more than one species or strain, and 95 of them (58%) showed various degrees of amino acid synthesis phenotype variability (Appendix A). The highest levels of variability (NVP > 5; OPVS > 2) were noted for the *Anaerococcus*, *Anaerotruncus*, *Coprococcus*, *Dialister*, *Lactobacillus*, *Lactococcus*, *Streptococcus*, and *Clostridium* genera from the Firmicutes phylum, a few genera from the Actinobacteria phylum, as well as the *Fusobacterium*, *Helicobacter*, *Prevotella* and *Sutterella* genera from other HGM phyla (Appendix A). At higher taxonomic ranks, the variability of amino acid synthesis phenotypes gradually increased, with the corresponding HGM families, namely Lactobacillales, Tissierellales, Bacteroidales and Actinomycetales, showing the highest OPVS values (Appendix A). Finally, we calculated the cumulative amino acid-specific variability scores across all analyzed taxa. Tryptophan biosynthesis appears to be the most variable amino acid synthesis phenotype at all taxonomic levels, followed by histidine, lysine and leucine biosynthesis that were also highly variable at the genus level. 

### 3.6. Profiling of Amino Acid Metabolic Potential of the Human Gut Microbiome 

To assess the ability of HGM bacterial communities to produce amino acids, we analyzed public metagenomic datasets using the previously developed phenotype profiling approach [25] and the obtained in this work Binary Phenotype Matrix (BPM) describing the predicted amino acid synthesis capabilities of reference HGM genomes. We calculated the CPIs for three large 16S rRNA datasets of fecal samples representing the human gut microbiome from the AGP [48], UKT [49], and the Tanzanian community of hunter-gatherers (Hadza) [51] (Appendix A). CPI values represent the expected fractions of respective amino acid synthesis phenotype carriers in the community. Their distributions across samples from each of the three datasets and for each amino acid synthesis phenotype demonstrate a similar pattern (Figure 4). 

We further analyzed PAD metrics of HGM samples, namely PAD_1 and PAD_0, that describe diversity for the sub-communities of phenotype carriers and non-carriers, respectively [47]. We computed the PAD_1 and PAD_0 values for samples from three analyzed 16S rRNA datasets and investigated their mutual dependence and link to CPI values (Appendix A). Amino acid synthesis phenotypes with the highest median CPI, such as chorismate, aspartate, glutamine, glycine, etc., reveal the largest PAD_1 and smallest PAD_0 values in each metagenomic dataset. To facilitate the comparison of metrics between distinct amino acid phenotypes, we analyzed the distributions of the relative CPI (rCPI) and the relative PAD (rPAD) values across samples in each dataset (Appendix A). For each amino acid, the rCPI was calculated as the ratio of the (1-CPI) value (community auxotrophy index) and CPI (community prototrophy index), while rPAD was defined as the ratio of PAD_0 (diversity of auxotrophs) and PAD_1 (diversity of prototrophs) for a given amino acid. The median rCPI and median rPAD values calculated for each amino acid phenotype and each dataset demonstrated mutual dependence (r^2^ = 0.81–0.91), where the highest rPAD and rCPI values correspond to tryptophan (Figure 5). This observation implies metabolic dependence between the abundance of bacterial prototrophs and community richness across the analyzed samples. Typically, rPAD values are 2–4 times larger than rCPI, which indicates that auxotrophs have greater diversity per unit of abundance. For most amino acids, the median values of rCPI and rPAD do not follow any particular pattern across datasets. However, the rPAD metric for tryptophan demonstrates consistency across three datasets, which probably puts an upper limit for the value of alpha diversity of tryptophan auxotrophs in the human gut microbiome. Finally, despite the great variation of rCPI and rPAD values, the western microbiome cohorts (AGP and UKT) have a greater similarity in distribution of median values for both metrics compared to the rural microbiome cohort of the Hadza, which demonstrates higher rPAD values for all amino acids except tryptophan (Appendix A).

### 3.7. Comparison of Amino Acid Production Phenotypes and Pathway Abundances 

To assess the phenotype-based functional profiling approach, we compared the obtained CPI profiles with predicted abundances of amino acid synthesis pathways in the AGP dataset obtained using PICRUSt2, a popular functional prediction tool for 16S rRNA metagenomic data [53]. First, we used the obtained in this work BPM describing amino acid synthesis as a custom input of functional traits in PICRUSt2, resulting in the calculation of relative phenotype abundances (RPA) for each amino acid in each AGP sample (Appendix A). The obtained PICRUSt2-based RPA values demonstrate high correlation coefficients with the CPI values for corresponding amino acids (Table 3). We further compared the PICRUSt2-predicted RPA profiles with relative abundance of major amino acid synthesis pathways in MetaCyc obtained using the default PICRUSt2 pipeline (Appendix A). For many amino acid biosynthesis pathways, the MetaCyc database [54] includes multiple alternative pathway variants (Table 3). For example, lysine biosynthesis is represented by four alternative MetaCyc pathways, two of which show high correlations with predicted RPA values for lysine, while two other pathway variants did not correlate with RPA. The highest correlation coefficients between RPA and relative pathway abundances were observed for the longest amino acid synthesis pathways, namely histidine and tryptophan, suggesting that an RPA-based approach for predictive functional profiling is preferable for short metabolic pathways. 

Finally, we analyzed the distribution of genes encoding amino acid biosynthetic enzymes in WGS metagenomic samples from the IBD study [56]. The TMM-normalized cumulative gene abundances for functional roles from seven analyzed biosynthetic pathways demonstrated uneven distribution across 384 WGS samples (Appendix A). For each analyzed pathway, we further selected two signature functional roles, corresponding to genes that are present in a single copy across the majority of amino acid prototrophic species in the reference HGM genomic collection (Appendix A). As a result, the abundance of each amino acid pathway was estimated as a sum of gene abundances for corresponding selected signature functional roles (Figure 6). Interestingly, the highest mean pathway abundances were observed for arginine, leucine, and cysteine, while histidine and other amino acids demonstrate lower mean pathway abundance values. We further obtained taxonomic profiles for WGS samples from the IBD dataset using the established Kraken 2/Bracken approach. The obtained NCBI taxonomies were mapped to the reference collection of 2856 HGM genomes to calculate the CPI indices for predicted relative abundances of amino acid producers in each sample (Appendix A). The CPI indices were compared with the calculated abundances of each respective amino acid synthesis pathway (based on a total abundance of genes from two selected signature functional roles). Fairly high Pearson correlation coefficients were obtained for tryptophan (0.66), followed by histidine (0.5) and arginine (0.49), while other amino acids demonstrated weak correlation coefficients (0.25–0.36) that can be explained by their high mean CPI values (>0.95) and by generally shorter metabolic pathways. 

## 4. Discussion

Microbial communities in the human gut are capable of producing a large number of biologically relevant metabolites including amino acids and their derivatives. Concentrations of proteinogenic amino acids in the large intestine are dependent on the following major processes: (i) de novo amino acid biosynthesis by HGM bacteria; (ii) dietary protein degradation by host proteases; (iii) the fermentation and transformation of free amino acids by HGM-expressed catabolic pathways and amino acid-active enzymes (e.g., generation of bioactive compounds from tryptophan); and (iv) host absorption [1,4,11,79]. HGM vary taxonomically and functionally from person to person, and also with time, being dependent on environmental (diet, exposure to pathogens, antibiotics, etc.), lifestyle (exercises, stress, tobacco and alcohol), and personal (genetics) factors [80]. Bioinformatics approaches for predictive functional profiling are important for the analysis of changes in metabolic function between various HGM samples. Previously, we developed a new approach of metabolic phenotype profiling to quantify the fractional representation of predicted metabolic features (phenotypes) and applied it for the analysis of B vitamin requirements in HGM metagenomic samples [25,26]. Here, we expanded this approach toward predictive functional profiling of amino acid requirements in HGM.

The subsystem approach to genome annotation and metabolic reconstruction techniques allow one to map known enzymes to biochemical pathways in model species, propagate the experimentally confirmed functional annotations to other genomes, and identify candidates for missing genes in pathway gaps [17,19,37]. Substantial phylogenetic and metabolic diversity of bacteria inhabiting the large intestine is a main factor contributing to numerous gaps in our understanding of amino acid biosynthesis in HGM species beyond model organisms such as *E. coli* and a few others [81]. In this study, we performed the subsystem-based metabolic reconstruction to predict the amino acid biosynthetic potential of 2856 reference HGM genomes. The reconstructed pathways include several alternative pathway variants (see Figure 1 and Table 1) and over 100 alternative enzymes represented by non-orthologous gene displacements and catalyzing 33 biochemical reactions (Appendix A). A few novel alternative enzymes allowed us to close the gaps in the analyzed biosynthetic pathway for the serine, arginine, threonine, and lysine pathways in a large number of HGM genomes (Table 2). Despite our efforts, a few metabolic pathways are still incomplete in many HGM genomes (see P* variants in Table 1). The most notable incomplete pathway variants are due to the absence of (i) phosphoserine phosphatase SerB in 450 genomes, and (ii) homoserine kinase ThrB in 50 genomes. Based on the identification of multiple alternative enzymes for each of these steps, and the presence of all other essential enzymes in the serine and threonine biosynthetic pathways of corresponding genomes, we tentatively assigned them prototrophic phenotypes for serine and threonine, respectively. We also identified the subsets of amino acid auxotrophs encoding partially truncated biosynthetic pathway variants and proposed that corresponding species not only have specific amino acid growth requirements but are also potentially capable of salvaging amino acid precursors such as citrulline (for Arg), meso-2,6-diaminopimelate (for Lys), shikimate (for chorismate), and anthranilate (for Trp). In silico predicted amino acid prototrophic and auxotrophic phenotypes were supported by published experimental data on nutritional requirements and production capabilities for amino acids (Appendix A). 

To apply the predicted amino acid prototrophic and auxotrophic phenotypes for quantitative analysis of community-wide amino acid production, we have converted them to a simple BPM (1/0) populated by 2856 reference HGM genomes. The obtained BPM contains a large number of prototrophic species that are capable of synthesizing most of the essential amino acids, while auxotrophic phenotypes were attributed to a small subset of phylogenetically diverse HGM species (Figure 3). To determine the potential limitations of phylotype-to-phenotype mapping used for the functional profiling of 16S metagenomic samples, we determined the level of intra- and interspecies variation of binary amino acid phenotypes in the reference set of HGM genomes. The observed variability of amino acid phenotypes can be explained by strain-level variations in the presence of corresponding amino acid biosynthetic genes, which are often organized into operons and can be subject to horizontal gene transfer or strain-specific gene loss events. As a result, we confirmed that most of the amino acid synthesis phenotypes are conserved between strains of the same species, while a small subset of species demonstrates a few variations in the presence of biosynthetic pathways for several amino acids, most notably Trp, Tyr, Lys, Ser, and Pro. We concluded that our current phylogeny-based phenotype mapping approach is robust for amino acid synthesis phenotypes with mean CPI prediction uncertainty values below 5%, thus enabling reliable predictive functional profiling for HGM metagenomic samples.

We further conducted metabolic phenotype profiling of 16S rRNA samples from various HGM metagenomic studies, including two large cohorts from the USA (AGP) and UK (UKT). To describe the metabolic capabilities of 16S samples, we utilized two previously introduced by us metrics: (i) CPI, which represents the expected fraction of phenotype carriers, i.e., bacterial cells with a particular metabolic capability (e.g., amino acid prototrophy); and (ii) PAD, which reflects the diversity of phenotype carriers (and non-carriers) in a microbiome sample. The distributions of CPI values for three analyzed datasets demonstrate a high degree of similarity and likely depict the optimal proportions between prototrophs and auxotrophs in the community for each amino acid. The highest mean CPI values corresponding to ~100% of prototrophs were observed for chorismate, glycine, glutamate, glutamine, and aspartate, while the lowest mean CPI was detected for tryptophan (0.65–0.80), more resembling the average frequency of phenotypes for the synthesis of many B-vitamins previously determined for the AGP dataset [25]. The remaining amino acids demonstrated somewhat high mean CPI values (0.90–0.97). The observed minor differences can be attributed to the dietary habits and other factors such as usage of antibiotics, probiotics, etc., which directly affect the species abundance of the luminal microbiota. A substantial level of tryptophan auxotrophy is a characteristic feature of all analyzed HGM samples that can be attributed to the following peculiarities about this amino acid: (i) tryptophan is encoded by a single UGG triplet, it has a rare protein occurrence and the highest molecular weight [82]; (ii) tryptophan is the most energetically expensive amino acid (its average biosynthetic cost is nearly 2–3 fold higher compare to all other amino acids except phenylalanine and tyrosine) [83]; (iii) tryptophan serves as a precursor for the synthesis of a range of neurological active metabolites including indoles, serotonin, kynurenine and tryptamine [84].

While CPI represents a fraction of phenotype carriers, such as a particular amino acid prototroph, the (1 − CPI) value describes the corresponding fraction of non-carriers (e.g., amino acid auxotrophs), and these two fractions sum up to 1. Unfortunately, PAD does not possess this additive property, namely, PAD_1 and PAD_0 do not sum up to total alpha diversity. To make a meaningful comparison between quantitative and qualitative characteristics for the sub-communities of each amino acid prototrophs and auxotrophs, we consider the corresponding ratios, i.e., rCPI = (1 − CPI)/CPI and rPAD = PAD_0/PAD_1. The scattered plot for medians of these two relative metrics calculated for all samples in each HGM dataset (Figure 5) demonstrates that the median rPAD is 2–4 higher than the median rCPI (depending on dataset and phenotype). This observation demonstrates that sub-communities of amino acid auxotrophs have a larger diversity per unit of abundance (diversity-driven) than the corresponding sub-communities of prototrophs (abundance-driven). For the majority of amino acids, the median values of rCPI and rPAD do not follow any particular pattern across datasets. However, for tryptophan, we observe that the rPAD metric demonstrates consistency across three datasets, which probably puts an upper limit for the value of alpha diversity of tryptophan auxotrophs in HGM samples. 

The phenotype-based functional profiling approach was also applied to WGS metagenomic samples from the IBD study, for which the metagenomic-based taxonomies were mapped to the reference genomes from BPM, in order to calculate their CPI profiles. Using our reference collection of reconstructed metabolic pathways, we calculated TMM-normalized gene abundances for each functional role in amino acid biosynthetic pathways. Comparison of the CPI profiles with the corresponding pathway gene abundance profiles in WGS samples revealed a generally good agreement for amino acid biosynthetic pathways with relatively high levels of auxotrophs, such as tryptophan. Finally, by using the existing predictive functional profiling approach, PICRUSt2, we compared the relative phenotype abundances determined using the amino acid synthesis BPM with the relative abundance of corresponding pathways in the MetaCyc database and reported a good agreement for many (but not all) amino acid synthesis pathways.

In conclusion, the subsystem-based approach combined with the comparative genomics analysis allowed us to reconstruct metabolic pathways and assign binary metabolic phenotypes for amino acid biosynthesis in the reference collection of HGM genomes. The obtained in this work BPM for amino acid production, as well as the previously reported BPM for the production of B-group vitamins [25,26], are useful reference datasets enabling the predictive functional profiling of microbial communities. In the near future, we plan a two-way expansion of these reference datasets to ensure: (i) a deeper coverage of HGM by propagation of the predicted binary phenotypes to a large number of HGM genomes and metagenome-assembled genomes (MAGs); and (ii) the inclusion of additional metabolic phenotypes to describe various nutrient utilization and fermentative end-product formation capabilities of HGM reference genomes. Application of our metabolic phenotype-based functional profiling approach to HGM metagenomic samples with rich metadata will have many practical applications in the diagnostics of various conditions and the development of personalized nutrition, as well as in a variety of in vivo and in vitro growth experiments with gut microbiota. 

## Figures and Tables

**Figure 1 microorganisms-10-00740-f001:**
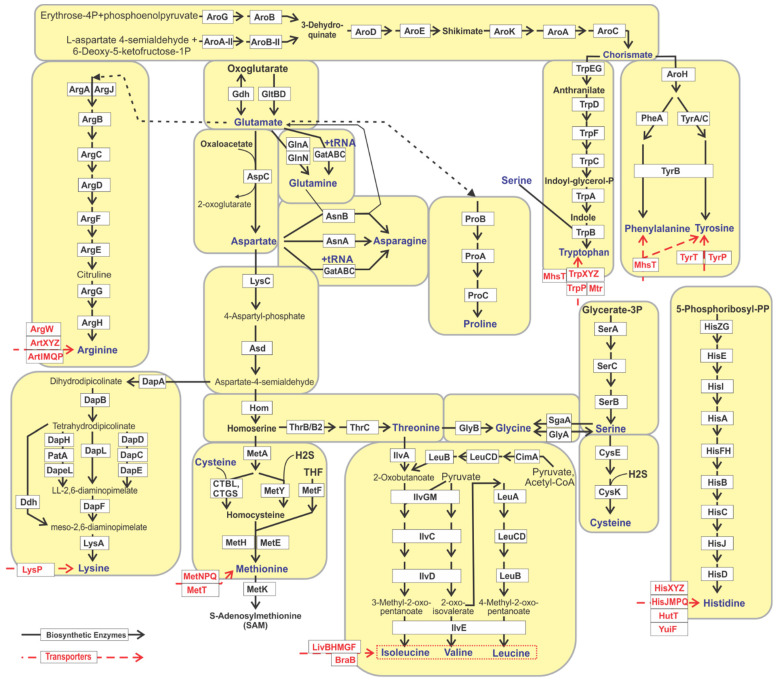
Reconstructed amino acid biosynthesis pathways in HGM genomes. Biosynthetic enzymes are shown in black using solid black arrows. Amino acid uptake transporters are shown in red using dashed red arrows.

**Figure 2 microorganisms-10-00740-f002:**
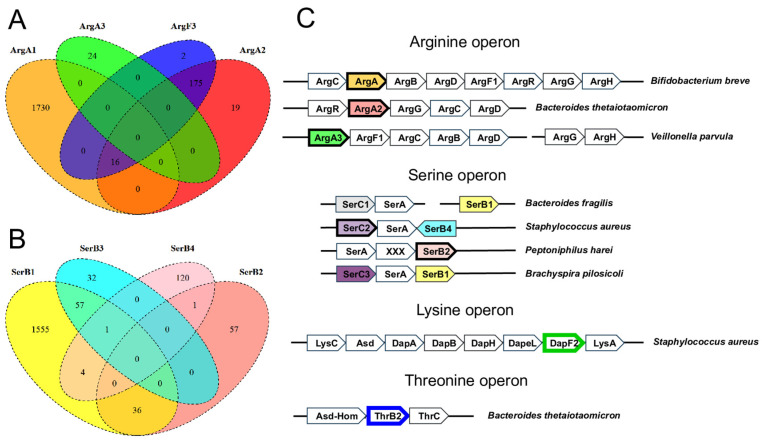
Genomic context of new non-orthologous enzymes in amino acid biosynthetic pathways. The Venn diagrams show co-occurrence of 4 different variants of N-acetylglutamate synthase ArgA (**A**), and phosphoserine phosphatase SerB (**B**). Gene co-localization is shown for selected genomes only (**C**).

**Figure 3 microorganisms-10-00740-f003:**
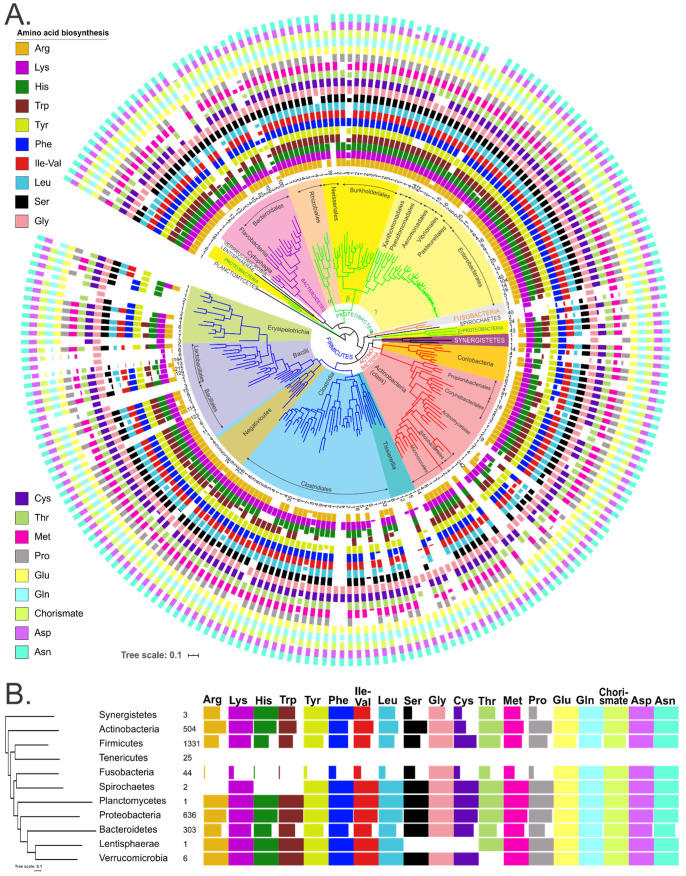
(**A**) Distribution of amino acids producers among analyzed HGM strains. Colored bars show the average amino acid prototrophy of each genus; empty bars represent auxotrophy. The phylogenetic tree of HGM genera was obtained using concatenated ribosomal proteins as previously described [25]. Number of analyzed strains per genus is shown in the inner circle. (**B**) Distribution of the amino acid producers at the phylum level. The number of analyzed genomes in each phylum is shown.

**Figure 4 microorganisms-10-00740-f004:**
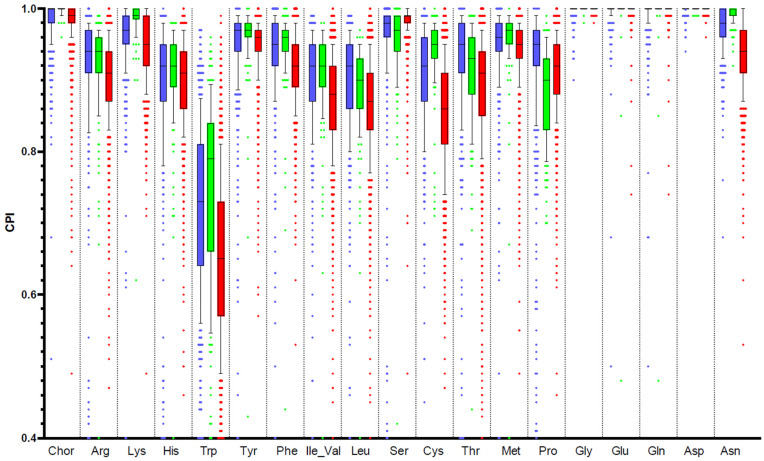
Distribution of Community Phenotype Indices (CPIs) for amino acid biosynthesis pathways in HGM samples. Samples from the AGP, UKT, and Hadza datasets are shown in blue, red, and green, respectively.

**Figure 5 microorganisms-10-00740-f005:**
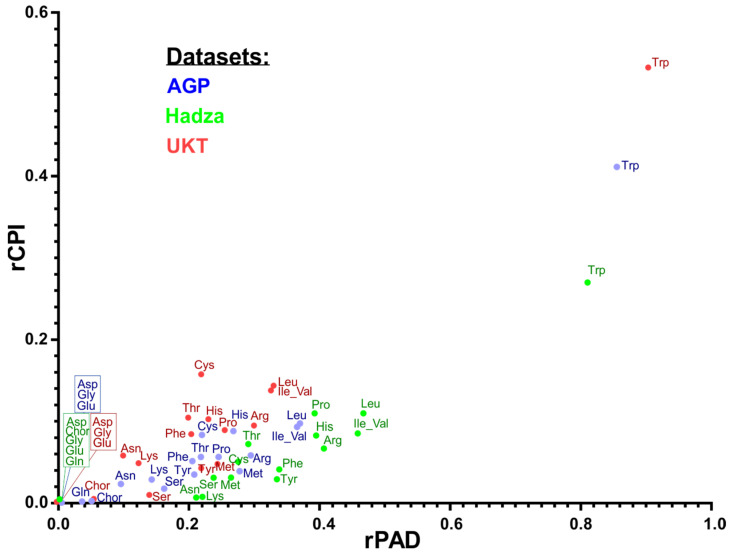
Mutual dependence of median rCPI and rPAD values for each amino acid synthesis phenotype in three HGM datasets.

**Figure 6 microorganisms-10-00740-f006:**
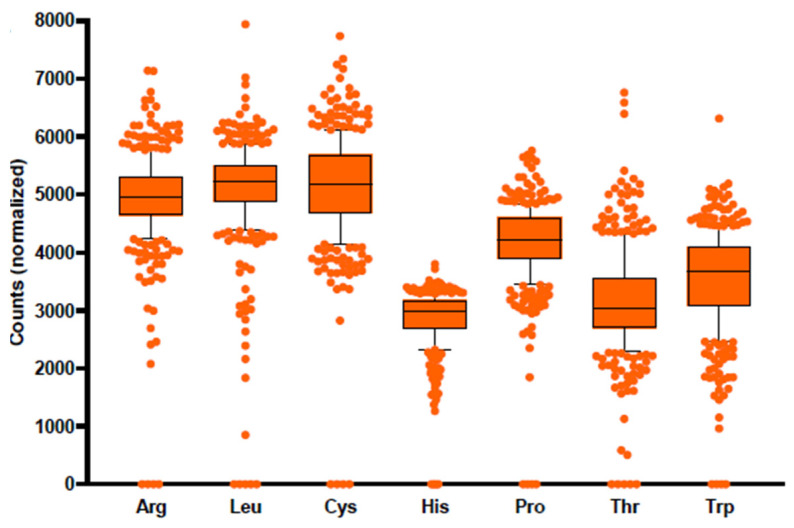
Distribution of metagenomic abundances for amino acid biosynthetic pathways in HGM samples from IBD dataset. Pathway abundances were calculated as a sum of TMM-normalized counts for two selected signature genes in each biosynthetic pathway (see Appendix A).

**Table 1 microorganisms-10-00740-t001:** In silico reconstruction and phenotype prediction for amino acid biosynthesis in HGM reference genomes.

Amino Acid	Pathway Signature and Biosynthetic a.a. Dependencies ^1^	PV ^2^	BP ^3^	No. Gen. ^4^	Growth Requirements ^5^
Proline	ProA, ProB	P	**1**	2275	--
	-	A	**0**	581	Pro
Threonine	Hom, ThrB, ThrC	P	**1**	2483	--
	Hom, ThrC	P*	**1**	50	(missing ThrB)
	-	A	**0**	323	Thr
Glycine	[GlyA/SgaA + Serine] or [GlyB + Threonine]	P	**1**	2182	--
	-	A	**0**	58	Gly
	GlyA/SgaA (no Serine)	P1	**1**	395	Ser, Gly
	GlyB (no Threonine)	P2	**1**	25	Thr, Gly
	[GlyA/SgaA (no Serine)] and [GlyB + (no Threonine)]	P3	**1**	23	Thr, Ser, Gly
Serine	SerA, SerC, SerB	P	**1**	1863	--
	SerA, SerC	P*	**1**	450	(missing SerB)
	-	A	**0**	543	Ser
Leucine & Isoleucine & Valine	IlvA, IlvG, (IlvM), IlvC, IlvD, (IlvE), LeuA, LeuB, LeuC, LeuD	P1	**1**	1912	--
	CimA, IlvG, (IlvM), IlvC, IlvD, (IlvE), LeuA, LeuB, LeuC, LeuD	P2	**1**	317	--
	IlvG, (IlvM), IlvC, IlvD, (IlvE), LeuA, LeuB, LeuC, LeuD	P*	**1**	17	(missing IlvA/CimA)
	IlvA, IlvG, (IlvM), IlvC, IlvD, (IlvE)	P3	**1/0**	60	Leu
	LeuA, LeuB, LeuC, LeuD, IlvE	A*	**0**	20	Ile, Val
	-	A	**0**	699	Leu, Ile, Val
Cysteine	CysE, CysK + Serine	P	**1**	1919	--
	-	A	**0**	526	Cys
	CysE, CysK (no Serine)	P1	**1**	411	Ser, Cys
Methionine	Hom, MetA, [CTBL, CTGS]/MetY, [MetH/MetE], (MetF), MetK	P	**1**	2252	--
	[MetH/MetE], (MetF), MetK	A1	**0**	83	Met
	Hom, MetA, [CTBL, CTGS]/MetY, MetK	A2	**0**	25	Met (missing MetH/E)
	MetK	A	**0**	496	Met
Lysine	LysC, Asd, DapA, DapB, DapH, (PatA), DapeL, DapF, LysA	P1	**1**	860	--
	LysC, Asd, DapA, DapB, DapD, (DapC), DapE, DapF, LysA	P2	**1**	1051	--
	LysC, Asd, DapA, DapB, DapL, DapF, LysA	P3	**1**	546	--
	LysC, Asd, DapA, DapB, Ddh, LysA	P4	**1**	643	--
	LysC, Asd, DapA, DapB, DapF, LysA	P*	**1**	38	(no amination pathway)
	LysA	A1	**0**	14	Lys, DAP
	-	A	**0**	248	Lys
Histidine	HisG, (HisZ), (HisE), HisI, HisA, HisH, HisF, HisB, HisD, (HisN), (HisC)	P	**1**	2097	--
	-	A	**0**	759	His
Tyrosine & Phenylalanine	PheA, TyrA/TyrC, (AroH), (TyrB)	FY	**1**	2257	--
	PheA, (AroH), (TyrB)	FA	**1/0**	40	Tyr
	TyrA/TyrC, (AroH), (TyrB)	AY	**0/1**	179	Phe
	-	AA	**0**	380	Tyr, Phe
Tryptophan	TrpA, TrpB, TrpC, (TrpD), TrpF, TrpEG	P	**1**	1810	--
	TrpA, TrpB, TrpC, (TrpD), TrpF, TrpEG (no Serine)	P1	**1**	140	Ser
	TrpA, TrpB	A1	**0**	14	Trp, indole precursors
	TrpA, TrpB, TrpC	A2	**0**	88	Trp, indole precursors
	TrpA, TrpB, TrpC, TrpD, TrpF	A3	**0**	38	Trp, anthranilate
	-	A	**0**	766	Trp
Arginine	(ArgA/ArgJ), (ArgB), ArgC, ArgD, ArgF, (ArgE), ArgG, ArgH	P	**1**	2061	--
	ArgG, ArgH	A1	**0**	251	Arg, citrulline
	ArgA, ArgB, ArgC, ArgD, (ArgF), (ArgE)	A2	**0**	5	Arg
	-	A	**0**	539	Arg
Chorismate	[AroG, AroB]/[AroA-II, AroB-II], AroD, AroE, AroK, AroA, AroC	P	**1**	2525	--
	AroD, AroE, AroK, AroA, AroC	P1	**1**	14	(missing AroG/AroB)
	AroG, AroB, AroD, AroE, AroK, (AroC)	P2	**1**	37	(missing AroA)
	AroG, AroB, AroD, AroE, AroA, AroC	P3	**1**	69	(missing AroK)
	AroG, AroB, AroD, AroK, AroA, AroC	P4	**1**	30	(missing AroE)
	AroK, AroA, AroC	As	**0**	13	Chorismate, shikimate
	-	A	**0**	169	Chorismate
Aspartate & Asparagine	AspC, AsnA/AsnB, (GatABC)	DN	**1**	1852	--
	AspC, GatABC	DAG ^	**1**	899	--
	AspC	DA	**1/0**	28	Asn
	(AsnB/AsnA), (GatABC)	AA	**0**	77	Asp, Asn
Glutamate & Glutamine	GltBD/Gdh, GlnA, (GatABC)	EQ	**1**	2663	--
	GltBD/Gdh, GatABC	EAG ^	**1**	17	--
	GltBD/Gdh	EA	**1/0**	20	Gln
	(GlnA), (GatABC)	AA	**0**	156	Gln, Glu

^1^ Special characters used in pathway signatures: Parenthesis denote functional roles that are not required to be present corresponding to enzymes that were not detected in all prototrophs. ‘/’ denotes alternative enzymes with the same functional role (at least one of these is required to be present). Biosynthetic dependencies on other amino acids (biochemical precursors) are shown in red, where parenthesis denote a respective amino acid growth requirement. ^2^ Pathway Variants: Asterisk denotes incomplete pathways with one or two essential enzymes missing. ‘^’ denote the presence of GatABC amidotransferase (see text). ^3^ Binary Phenotypes: ‘0’ and ‘1’ correspond to auxotrophs and prototrophs, respectively. ^4^ Number of genomes possessing a pathway variant. ^5^ ‘--’ denotes no growth requirement in predicted prototrophs; comments in parenthesis describe missing biosynthetic enzymes or pathways. DAP, meso-2,6-diaminopimelate.

**Table 2 microorganisms-10-00740-t002:** Novel amino acid biosynthesis enzymes predicted as non-orthologous gene displacements in HGM genomes.

Pathway	Enzyme	Predicted Functional Role	Occurrence ^1^	Evidence ^2^	Example ID ^3^
Arginine	ArgA2	N-succinylglutamate synthase (EC 2.3.1.-)	7.4%	CO, CL, CR, CF	Q8A1A5
	ArgA3	N-acetylglutamate synthase (EC 2.3.1.1)	0.8%	CO, CL	W3Y6L2
Serine	SerC2	Phosphoserine aminotransferase (EC 2.6.1.52)	4.0%	CO, CF, CL	Q2FXK2
	SerC3	Phosphoserine aminotransferase (EC 2.6.1.52)	6.1%	CF, CL	A5I0W7
	SerB2	Phosphoserine phosphatase (EC 3.1.3.3)	3.6%	CO, CF	C4IFQ5
Threonine	ThrB2	Homoserine kinase (EC 2.7.1.39)	10.6%	CO, CL	Q5LHR7
Lysine	DapF2	Diaminopimelate epimerase (EC 5.1.1.7)	4.4%	CO, CL, CR	W1W731

^1^ Percentage of HGM genomes possessing a non-orthologous enzyme. ^2^ Genome context evidence: CO, co-occurrence; CL, co-localization; CR, co-regulation; CF, common functional class. ^3^ Uniprot protein ID.

**Table 3 microorganisms-10-00740-t003:** Pearson’s correlation coefficient for comparison of CPIs, relative phenotype abundance (RPA) and relative pathway abundance (MetaCyc) for amino acid synthesis pathways obtained by PICRUST2 pipeline on HGM samples from the AGP dataset.

RPA vs. CPI	RPA vs. MetaCyc	MetCyc Pathway Name and Annotation
0.86	0.54	ARGSYNBSUB-PWY: L-arginine biosynthesis II (acetyl cycle)
	0.59	ARGSYN-PWY: L-arginine biosynthesis I (via L-ornithine)
	0.31	PWY-5154: L-arginine biosynthesis III (via N-acetyl-L-citrulline)
	0.60	PWY-7400: L-arginine biosynthesis IV (archaebacteria)
0.81	0.92	HISTSYN-PWY: L-histidine biosynthesis
0.60	0.64	ILEUSYN-PWY: L-isoleucine biosynthesis I (from threonine)
	0.71	PWY-5101: L-isoleucine biosynthesis II
	0.69	PWY-5103: L-isoleucine biosynthesis III
	0.56	PWY-5104: L-isoleucine biosynthesis IV
0.67	0.70	LEUSYN-PWY: L-leucine biosynthesis
0.69	0.27	DAPLYSINESYN-PWY: L-lysine biosynthesis I
	−0.05	PWY-2941: L-lysine biosynthesis II
	0.71	PWY-2942: L-lysine biosynthesis III
	0.72	PWY-5097: L-lysine biosynthesis VI
0.71	−0.16	HOMOSER-METSYN-PWY: L-methionine biosynthesis I
	−0.17	HSERMETANA-PWY: L-methionine biosynthesis III
0.74	0.86	TRPSYN-PWY: L-tryptophan biosynthesis
0.60	0.64	VALSYN-PWY: L-valine biosynthesis

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
