# Peer review of "Genomics-Based Reconstruction and Predictive Profiling of Amino Acid Biosynthesis in the Human Gut Microbiome"

_microorganisms, 2022, doi:10.3390/microorganisms10040740_

Round 1
Reviewer 1 Report
The manuscript from Ashniev et al. analyses about 2,856 genomes representing 823 species with the goal to reconstruct the amino acids metabolic pathways.
The work is not easy to follow and often it is not clear what is the goal. If understood correctly, the authors first analyze 2,856 genomes to derive a set of 283 functional roles.
Then it is not clear how these results were used to analyze the American Gut Project, the TwinsUK, and the Hazda cohorts. The three cohorts have 16S microbiome profiles, and PICRUSt2 was used to infer the functional profiles. Now, the PICRUSt2 profiles were used to compare the CPI index, which seems to be derived from 16S data as well.
Wouldn't be worth comparing the functional annotations the authors derived with functional potential profiles computed from shotgun metagenomics data? This would be more important because functional profiles from 16S can be very noisy.
Also, from the three cohorts some samples were excluded, based on "we selected only samples that have abundance coverage >75% and number of unclassified species <10%.", this is not clear what it means, please clarify.
Typos:
- Abstract: "varinats" --> "variants"
Author Response
The revised manuscript was improved to address all points in the following way:
Point 1: The work is not easy to follow and often it is not clear what is the goal. If understood correctly, the authors first analyze 2,856 genomes to derive a set of 283 functional roles.
Response 1: Absolutely! We revised both the introduction, methods and results to improve the rational and logics of our study. You are correct, the first two-thirds of the study deal with genomics-based metabolic recontruction of target biochemical pathways in the reference set of microbial genomes from the human gut. This analysis is presented in substantial details that include (i) the genome context analysis of incomplete pathway variants; (ii) identification of new potential enzyme variants in pathway with existing missing genes; (iii) classification of the studied bacteria by their prototrophic vs. auxotrophic phenotypes (for each amnio acid); (iv) a cross-species comparison to assess the extent of conservation of the assigned binary metabolic phenotypes at distinct taxonomic levels; (v) validation of prototrophic & auxotrophic phenotypes through the literature.
Point 2: Then it is not clear how these results were used to analyze the American Gut Project, the TwinsUK, and the Hazda cohorts. The three cohorts have 16S microbiome profiles, and PICRUSt2 was used to infer the functional profiles. Now, the PICRUSt2 profiles were used to compare the CPI index, which seems to be derived from 16S data as well.
Response 2: Yes, we first obtained the taxonomic profiles for each of these 3 datasets, and then computed CPI indices for each sample and each amino acid pathway using the binary phenotype matrix for reference genomes. The obtained CPI indices represent expected fraction of microbial cells possessing a particular amino acid biosynthetic phenotype. The obtained distribution of CPI indices were then compared with the most advanced existing approach, PICRUSt2, predicting pathway abundance via a combination of MinPath algorithm with MetaCyc pathways. The relative pathway abundances obtained using PICRUSt2 revealed a generally good agreement for most (but not all) amino acid phenotypes. The highest correlation coefficients were observed for longest amino acid synthesis pathways (histidine and tryptophan). The substantial differences between CPI pathway abundances observed for short biosynthetic pathways and pathways with several alternative routes could be partially explained by the differences in the applied rules and in the extent of
gene/pathway curation. We expnded the respective section in the results and also added a discussion to cover these findings. We added these results to the section 3.7 and also provided an additional Table S8 and FIgure 6. In this section we analysed the distribution of genes encoding amino acid biosynthetic enzymes in WGS metagenomic samples from the IBD study
Point 3: Wouldn't be worth comparing the functional annotations the authors derived with functional potential profiles computed from shotgun metagenomics data? This would be more important because functional profiles from 16S can be very noisy.
Response 3: We agree with this point and decided to expand our analysis to calculate the functional potential of shotgun metagenomic samples (WGS). We analyzed the TMM-normalized abundances of genes encoding amino acid biosynthetic enzymes in 384 WGS samples from the IBD study. For each analyzed pathway, we further selected two signature functional roles, corresponding to genes that are present in a single copy across the majority of amino acid prototrophic species in the reference HGM genomic collection. We further obtained taxonomic profiles for WGS samples from the IBD dataset using the established Kraken 2/Bracken approach and calculated CPI indices for predicted relative abundances of amino acid producers in each sample. The CPI indices were compared with calculated abundances of each respective amino acid synthesis pathway. Comparison of the CPI profiles with the corresponding pathway gene abundance profiles in WGS samples revealed a generally good agreement for amino acid biosynthetic pathways with relatively high leveles of auxotrophs, such as tryptophan.
Point 4: Also, from the three cohorts some samples were excluded, based on "we selected only samples that have abundance coverage >75% and number of unclassified species <10%.", this is not clear what it means, please clarify.
Response 4: The sample filtering procedure is now described in great details in Methods (lines 174-178). The metabolic phenotype profiling pipeline consists of several steps, first of which establishes a map between the analyzed ASVs (from 16S samples) and the reference organisms in the BPM based on the 16S rRNA nucleotide identity. ASVs with high nucleotide identity (greater than 0.9) were considered “mapped”, while the other ASVs were considered as “non-mapped” and discarded. Samples with less than 75% abundance coverage (i.e., the abundance of “mapped” ASVs) were discarded.
Final remarks: all typos and grammatic errors were corrected throught the manuscript.
Reviewer 2 Report
In this study, authors have analyzed>2,800 genomes to reconstruct amino acid metabolism phenotypes and classified them as prototrophic vs auxotrophic. The authors have used established tools to identify new non-orthologous enzyme displacements in amino acid biosynthetic pathways. Authors also related predicted phenotype findings to previously published experimental data to signify their method is with 87% of agreement with auxotrophic. Overall, the presented study is well designed and interesting to a broad microbiome research community.
Check minor typos, and misspellings throughout the manuscript.
Author Response
Thank you! We proof checked the entire manuscript and corrected all typos and grammatic errors.
Reviewer 3 Report
The authors described a very interesting issue. From a microbiological point of view, this is a relatively new approach to analyzing the microbiome.
The conclusion contains a few speculative observations. In my opinion, a better solution would be to present the results first, and the discussion, along with most of the statements that are in the conclusion, in a separate paragraph. The conclusion should contain a few short conclusions.
I have doubts whether the conclusions and suggestions contained in the conclusion (e.g. lines 618-621) are valid if only the modeling data were analyzed and not the metadata concerning not only the composition of the microbiota, but also the lifestyle, diet and accompanying diseases of the given population.
Author Response
The revised manuscript was improved to add the results of new data analysis (WGS samples, section 3.7), revised the methods and results sections and to add the discussion section with brief conclusions at the end. We address the specific points in the following way:
Point 1: The conclusion contains a few speculative observations. In my opinion, a better solution would be to present the results first, and the discussion, along with most of the statements that are in the conclusion, in a separate paragraph. The conclusion should contain a few short conclusions.
Response 1: Thatn you for your recommendation! We removed the previously presented Concusion section and wrote a more extensive Discussion section, with the last paragraph that summarize in a short way the most important conclusions of this study.
Point 2: I have doubts whether the conclusions and suggestions contained in the conclusion (e.g. lines 618-621) are valid if only the modeling data were analyzed and not the metadata concerning not only the composition of the microbiota, but also the lifestyle, diet and accompanying diseases of the given population.
Response 2: We agree that this statement was over speculative and was not directly related to the results of this study. Thus, we removed this statement from the manuscript.
Round 2
Reviewer 1 Report
I thank the authors for revising the manuscript addressing all raised comments.